# Mechanism of CK2.3, a Novel Mimetic Peptide of Bone Morphogenetic Protein Receptor Type IA, Mediated Osteogenesis

**DOI:** 10.3390/ijms20102500

**Published:** 2019-05-21

**Authors:** Vrathasha Vrathasha, Hilary Weidner, Anja Nohe

**Affiliations:** Department of Biological Sciences, University of Delaware, Newark, DE 19716, USA; vrathash@udel.edu (V.V.); weidnerh@udel.edu (H.W.)

**Keywords:** bone, osteoporosis, osteogenesis, BMP2, CK2, CK2.3, osteogenic signal transduction, western blots, RT-qPCR

## Abstract

Background: Osteoporosis is a degenerative skeletal disease with a limited number of treatment options. CK2.3, a novel peptide, may be a potential therapeutic. It induces osteogenesis and bone formation in vitro and in vivo by acting downstream of BMPRIA through releasing CK2 from the receptor. However, the detailed signaling pathways, the time frame of signaling, and genes activated remain largely unknown. Methods: Using a newly developed fluorescent CK2.3 analog, specific inhibitors for the BMP signaling pathways, Western blot, and RT-qPCR, we determined the mechanism of CK2.3 in C2C12 cells. We then confirmed the results in primary BMSCs. Results: Using these methods, we showed that CK2.3 stimulation activated *OSX*, *ALP*, and *OCN*. CK2.3 stimulation induced time dependent release of CK2β from BMPRIA and concurrently CK2.3 colocalized with CK2α. Furthermore, CK2.3 induced BMP signaling depends on ERK1/2 and Smad1/5/8 signaling pathways. Conclusion: CK2.3 is a novel peptide that drives osteogenesis, and we detailed the molecular sequence of events that are triggered from the stimulation of CK2.3 until the induction of mineralization. This knowledge can be applied in the development of future therapeutics for osteoporosis.

## 1. Introduction

Osteoporosis is a skeletal disorder and is the leading cause of fractures in the USA. 10 million Americans are diagnosed with osteoporosis and another 44 million Americans are at an increased risk of a fracture due to their low bone density. In 2005, there were 2 million osteoporosis-related fractures that resulted in a total cost of 18 billion dollars, and this number is estimated to increase to 3 million fractures and cost 25.3 billion dollars by 2025 [1,2]. This is due to the rapid increase in the world’s aging populations, thus placing a substantial burden of disability and costs on the individuals, as well as on the society. The underlying cellular pathogenesis causing osteoporosis is due to: (a) excessive bone resorption by osteoclasts and/or (b) failure of osteoblasts to produce and replace the resorbed bone, leading to weak and porous bones [3,4,5,6]. The majority of the existing treatments for osteoporosis [4,7,8,9] such as bisphosphonates (e.g., alendronate, risedronate, and ibandronate) [10], selective estrogen receptor modulators (SERMs) (e.g., tamoxifen and raloxifene) [11,12], calcitonin, and denosumab, target inhibition of osteoclast activity. However, prolonged use of such treatments can prevent bone remodeling, thereby micro-damages and fractures in the bone may go unresolved. Teriparatide, a recombinant human- parathyroid hormone (rh-PTH), composed of 1–34 amino acid fragments of the parathyroid hormone (PTH) [13,14], is an anabolic treatment that addresses the issue of increasing bone mineral density (BMD) in osteoporotic patients, but excess concentration of serum PTH is known to cause hyperparathyroidism and inadvertently increases bone resorption and exacerbates the osteoporotic condition [15]. We developed a novel peptide, called CK2.3, that stimulates osteoblast differentiation and bone formation in vitro and in vivo [16,17,18,19,20]. However, CK2.3 not only increases osteoblast numbers, but simultaneously decreases the number of osteoclasts [16,18,20], making it a potential treatment that can possibly reverse the osteoporotic phenotype.

Bone morphogenetic proteins (BMPs) are members of the transforming growth factor-β (TGF-β) superfamily and some members, such as BMP2, are involved in intramembranous and endochondral ossification. They regulate mesenchymal stem cell (MSC) differentiation into osteoblasts [21,22,23], making BMPs a possible treatment for osteoporosis. In 2002, recombinant human bone morphogenetic protein 2 (rh-BMP2) was approved by the Food and Drug Administration (FDA) for human spinal surgery, but starting in 2006, many complications were cited from its usage [24]. As a result, it is quintessential to characterize BMP2-mediated signaling pathways leading to osteogenesis. BMP2 signals through serine/threonine kinase receptors. One of the well-known BMP2 signaling pathways occurs via bone morphogenetic protein receptor type IA (BMPRIA) and bone morphogenetic protein receptor type II (BMPRII). The role of BMPRIA in regulating osteoblast function is well established [25,26,27,28]. C2C12 cells (murine myoblasts) are often used as model cell line to study MSC differentiation into osteoblasts. C2C12 cells express BMPRIA [29] and the receptor is crucial in mediating BMP2-induced differentiation of C2C12 cells into osteoblasts [30]. We identified an interaction between protein kinase 2 (CK2) and BMPRIA and determined the site on BMPRIA between the amino acids 213–217 (SLKD) to be a potential CK2 phosphorylation site. Overexpression of BMPRIA mutant (214S-A) lacking the serine amino acid in the phosphorylation site leads to increased mineralization via the ERK-MEK signaling pathway [16,31]. Moreover, using the SLKD phosphorylation site of CK2 on BMPRIA, a blocking peptide called CK2.3 was developed and it functions by mimicking the CK2 binding site and prevents the interaction between CK2 and BMPRIA [19]. The knockdown of BMPRIA using siRNA completely abolished CK2.3-mediated osteogenesis, as seen using reporter gene assay [16]; thus, implying CK2.3 functions downstream of BMPRIA and independent of external BMP2 stimulus. Using CK2.3-Qdot^®^s as a fluorescent bio-conjugate, CK2.3 was observed to be internalized by caveolae mediated endocytosis starting at 6 h post-stimulation [32]. In vitro experiments conducted in C2C12 cells, primary calvarial osteoblasts, and bone marrow stromal cells (BMSCs) revealed that 100 nM of CK2.3 induces significant mineralization independent of BMP2 [16,17,18,19]. Furthermore, subcutaneous injection into the right side of the calvaria of 4-week-old female C57BL/6J mice [16] and systemic injection of CK2.3 in 8-week-old and 6-month-old female C57BL/6J mice [18,33] increases total bone area, areal bone mineral density (aBMD), and mineral apposition rate (MAR) compared to control and BMP2 injected mice, and also shows elevated levels of osteoblast markers and a reduced number of osteoclasts. Even though CK2.3 has osteogenic potency, much of the mechanism that leads to the mineralization of C2C12 cells and primary BMSCs remained elusive. Furthermore, if CK2.3 can be considered as a potential candidate for treatment of osteoporosis, the detailed molecular events implicated in the differentiation of MSCs into osteoblasts by CK2.3 must be delineated.

Here, we investigated the mechanism of CK2.3 within C2C12 cells. The commitment to osteoblastic lineage involves the expression of *RUNX2* and *Osterix (OSX)* gene. *RUNX2* is regarded as the master regulator of osteogenesis. *OSX* is a zinc-finger-containing transcription factor which was first discovered in C2C12 cells and acts downstream of BMP-mediated osteogenic activity. Some of the markers used to identify osteoblasts are alkaline phosphatase (ALP) and osteocalcin (OCN). ALP is a membrane-bound glycoprotein that is heavily expressed in mineralized tissue and it is reported to function by increasing the concentration of inorganic phosphate, which is a major component of the bone matrix [34,35,36]. Osteocalcin is a hormone specifically secreted by osteoblasts and the Gla residues present in the osteocalcin are known to have a high affinity for binding to hydroxyapatite crystals [37,38]. Osteocalcin is the most abundant non-collagenous protein present in the bone matrix, and it is essential for bone metabolism [39]. We determined the time-dependent upregulation of osteoblast specific genes such as *RUNX2, OSX, ALP,* and *OCN* using RT-qPCR, the effect of CK2.3 on the interaction between CK2 and BMPRIA, and whether CK2.3 mediated osteogenesis functions via the Smad-dependent or Smad-independent signaling pathways.

Stimulation of C2C12 cells with 100 nM of CK2.3 induced a time-dependent release of CK2β from BMPRIA. Additionally, we confirmed a gradual increase in co-localization between CK2α and CK2.3 starting at 6 h, 12 h, and 18 h post-stimulation using CK2.3-Qdot^®^s. We determined that CK2.3 activated the BMPRIA downstream signaling pathways such as Smad1/5/8, ERK1/2, and Akt1/2/3. However, using pharmacological inhibitors such as U0126-EtOH, SB202190, and MK-2206 2HCl against MEK1/2, p38 MAPK, and Akt1/2/3, respectively, and Smad4 siRNA to silence Smad1/5/8, we were able to identify that the ERK1/2 and Smad1/5/8 signaling pathways were critical for CK2.3-mediated osteogenesis. Current research focuses on Smad1/5/8-mediated osteogenesis and neglects the role of Smad-independent signaling pathways. Here, we report the ERK1/2 signaling pathway to be essential for osteogenesis by CK2.3; an alternate proposal to a commonly held dogma. The significance of ERK1/2 and Smad1/5/8 signaling pathways on CK2.3 mediated osteogenesis was also verified in primary BMSCs. Now that we are able to characterize the activity of CK2.3, it can be regarded as a potential candidate for the treatment of osteoporosis, subject to further research in advanced animal models, or at the very least, the information that we have gained from this research can be implemented in the development of future treatments for osteoporosis.

## 2. Results

### 2.1. CK2.3 Stimulation Led to the Upregulation of Osterix, Alkaline Phosphatase, and Osteocalcin Genes in C2C12 Cells

Differentiation of osteoblasts from pluripotent MSCs is a highly sequenced process that involves the expression of particular genes at specific stages of differentiation [40,41,42,43,44]. C2C12 cells were treated with 100 nM of CK2.3 from day 1–5. Total RNA was isolated at each respective time point and a two-step RT-PCR was performed. Expression of osteoblastic genes *RUNX2, OSX*, *ALP*, and *OCN* were analyzed using RT-qPCR. C2C12 cells stimulated with 100 nM of CK2.3 elevated the expression of *RUNX2* mRNA on day 4, however it was not statistically significant (Figure 1A). However, CK2.3 treatment resulted in a gradual increase in the expression of *OSX* starting from day 1 and was significantly elevated on day 2 and day 3 by 3.95 ± 0.66 and 4.1 ± 0.9 fold, respectively (Figure 1B). The expression pattern of *OSX* by CK2.3 is similar to the BMP2-mediated induction of *OSX* in C2C12 cells [45,46,47]. *ALP* expression was significantly upregulated on day 5 by 3.98 ± 0.8 fold, following treatment with 100 nM of CK2.3 (Figure 1C), the upregulation of *ALP* mRNA expression is similar to C2C12 cells stimulated with BMP2 [48,49]. It is reported that, as the osteoblast differentiation progresses, the expression of *OCN* gene is upregulated [34,49,50] and a similar result was observed in our experiment with CK2.3 in C2C12 cells, i.e., on day 5, *OCN* expression was significantly increased by 3.63 ± 1.36 fold (Figure 1D). *GAPDH* was used as the house-keeping gene.

### 2.2. Time-Dependent Release of CK2 from BMPRIA at 12 h and 18 h Post CK2.3 Stimulation 

The interaction between CK2.3 and BMPRIA must be determined in order to investigate the signaling pathways upregulated by CK2.3. We previously identified an association between CK2 and BMPRIA [19], however the time frame of interaction and release of CK2 were unknown. C2C12 cells were cultured and stimulated with 100 nM of CK2.3 for 6 h, 12 h, and 18 h. Cells were fixed, endogenous BMPRIA and CK2α were immuno-fluorescently labeled, and the co-localization of BMPRIA and CK2α were visualized using a confocal laser scanning microscope (Figure 2A). Co-localization of BMPRIA (green) and CK2α (red) was observed in unstimulated cells (Figure 2A(A)) and at 6 h post CK2.3 stimulation (Figure 2A(B)). However, with prolonged CK2.3 stimulation, decreased co-localization between CK2α and BMPRIA was observed at 12 h (Figure 2A(C)) and 18 h (Figure 2A(D)), respectively. This indicates that CK2.3 was mediating the release of CK2 from BMPRIA after 6 h. To confirm this finding, C2C12 cells were either left unstimulated or stimulated with 100 nM of CK2.3, lysed, and immuno-precipitated for BMPRIA. Detection of the association with CK2β was completed by SDS-PAGE followed by Western blot. There was a decreased amount of CK2β detected in stimulated cells when compared to unstimulated cells, with the maximum disassociation observed at 18 h (Figure 2B). This indicates and confirms that CK2 was fully released from BMPRIA at 18 h post CK2.3 stimulation. 

### 2.3. Time-Dependent Co-Localization of CK2 with CK2.3-Qdot^®^s at 6 h, 12 h, and 18 h Post-Stimulation

To determine the activity of CK2.3 in vitro, CK2.3 is conjugated to fluorescent nanoparticles called quantum dot^®^s (Qdot^®^s) to track CK2.3 within C2C12 cells. Qdot^®^s are semiconductor nanocrystals and their unique photophysical properties make them ideal for bio-imaging applications. The development and characterization of CK2.3-Qdot^®^s has been published [32].

C2C12 cells were either left unstimulated or stimulated with 100 nM of CK2.3-Qdot^®^s for 6 h, 12 h, and 18 h. After each respective time point, cells were fixed and stained for CK2α and nucleus of the cell. Images of the cell were taken using confocal laser scanning microscopy. At 6 h post-stimulation, slight co-localization between endogenous CK2α (red) and CK2.3-Qdot^®^s (green) (Figure 3B) was observed. However, the co-localization peaked at 12 h post-stimulation, as seen in the merged images at zoom-4 and zoom-10 (Figure 3C). At 18 h, we mostly observed the co-localization between CK2α and CK2.3-Qdot^®^s around the plasma membrane of the cell (Figure 3D). These data indicate that CK2.3 co-localized with protein kinase 2 intracellularly.

### 2.4. Smad1/5/8, ERK1/2, and Akt1/2/3 Signaling Pathways are Upregulated Following Stimulation with 100 nM of CK2.3 in C2C12 Cells

Some of the key signaling pathways involved in regulating MSC differentiation into osteoblasts are Smad1/5/8, ERK1/2, Akt1/2/3, and p38 MAPK. Smad1/5/8 constitutes the canonical signaling pathway [51,52,53,54], whereas ERK1/2, Akt1/2/3, and p38 MAPK are part of the non-canonical pathway [55,56,57,58,59,60,61,62] and are not well characterized. To determine the signaling pathway utilized by CK2.3 to induce osteoblast differentiation, C2C12 cells were either left unstimulated or stimulated with 100 nM of CK2.3 for 6 h, 12 h, and 18 h over days 1–5. After each time point, cells were lysed, protein was extracted and analyzed using SDS-PAGE, followed by Western blot to determine the presence of total and phosphorylated Smad1/5/8, ERK1/2, Akt1/2/3, and p38 MAPK signaling molecules. In the case of Smad1/5/8, CK2.3 gradually upregulated the expression of p-Smad1/5/8 and total-Smad1/5/8 compared to unstimulated cells over the course of 5 days (Figure 4A). With respect to ERK1/2 signaling, p-ERK and total ERK1/2 expression were upregulated (Figure 4B). Similarly, CK2.3 increased the expression of p-Akt1/2/3 (Figure 4C) compared to unstimulated cells. However, expression of p-p38 MAPK and total p38 MAPK stayed relatively constant through the CK2.3 treatment (Figure 4D). β-actin was used as the loading control to compare the expression of each signaling molecule.

### 2.5. Inhibition of ERK1/2 and Smad1/5/8 Signaling Pathways Led To Decreased Mineralization by CK2.3 in C2C12 Cells

To determine the pathway responsible for CK2.3 mediated osteogenesis, we used inhibitors such as U0126-EtOH, SB202190, and MKK-2206 2HCl against ERK1/2, p38 MAPK, and Akt1/2/3 signaling pathways, respectively. An optimal concentration of specific signaling inhibitors that resulted in significant reduction in mineralization was determined by performing von Kossa assay. Furthermore, to confirm that the reduction in mineralization was solely due to the inhibition of the signaling pathway and not due to reduced number of viable cells, viability of the cells were determined by MTT (3-(4,5-dimethylthiazol-2-yl)-2,5-diphenyltetrazoilum bromide) tetrazolium reduction assay and total number of cells were quantified using ImageJ analysis. Smad1/5/8 signaling pathway was silenced using Smad4 siRNA.

#### 2.5.1. 500 nM of ERK1/2 Inhibitor (U0126-EtOH) Induces Significant Reduction in Mineralization Without Affecting the Viability of C2C12 Cells

Von Kossa assay is a commonly used technique to detect calcium deposits [63]. C2C12 cells were either left unstimulated or stimulated with 500 nM, 1 μM, and 5 μM of ERK1/2 Inhibitor (U0126-EtOH), p38 MAPK inhibitor (SB202190), and Akt1/2/3 (MKK-2206 2HCl). Also, only in the case of p38 MAPK inhibitor, we used an additional concentration of 10 μM, as neither of the concentrations of SB202190 yielded a significant reduction in mineralization. After 24 h, the cells were then treated with 100 nM of CK2.3. Stimulation of cells with varying concentration of signaling inhibitors, followed by CK2.3 stimulation, was repeated for a total of two times over the course of the experiment. On the sixth day, cells were fixed, and calcium deposits were stained using von Kossa assay. The analysis of images of cells using ImageJ revealed that 500 nM of ERK1/2 inhibitor (U0126-EtOH) (Figure 5A), 5 μM of Akt inhibitor (MKK-2206 2HCl) (Figure 5B), and 10 μM of p38 MAPK inhibitor (SB202190) (Figure 5C) had a significant reduction in mineralization compared to cells only treated with 100 nM of CK2.3.

Viability of cells following stimulation with varying concentrations of pharmacological inhibitors and CK2.3 was determined by MTT reduction assay and quantification of total number of cells. C2C12 cells were either left unstimulated or stimulated with 500 nM of U0126-EtOH, 5 μM of MKK-2206 2HCl, and 10 μM of SB202190 to inhibit ERK1/2, Akt1/2/3, and p38 MAPK signaling pathways, respectively. The next day, the cells were then treated with 100 nM of CK2.3. Halfway through the experiment, the cells were once again treated with the respective concentrations of signaling inhibitors and 100 nM of CK2.3. In the end, viability of the cells was determined by conducting MTT assay. Our data show that 500 nM of ERK1/2 Inhibitor (U0126-EtOH) did not induce significant reduction in the number of viable cells compared to unstimulated cells or cells only treated with 100 nM of CK2.3, however, cells treated with 5 μM of Akt1/2/3 inhibitor (MKK-2206 2HCl) and 10 μM of p38 MAPK inhibitor (SB202190) resulted in a significant decrease in the number of viable cells (Figure 5D). This data was further verified by performing a cell counting experiment (Figure 5E). C2C12 cells that were either left unstimulated or stimulated with signaling inhibitors and CK2.3 were then fixed after 6 days. Images of cells were taken, and total number of cells were quantified using ImageJ analysis. Our data once again revealed that there was no significant difference amongst cells that were left untreated, cells that were only treated with 100 nM of CK2.3, or cells that were treated with 500 nM of ERK1/2 inhibitor (U0126-EtOH) and 100 nM of CK2.3; however, cells that were treated with 5 μM of Akt inhibitor (MKK-2206 2HCl) and 10 μM of p38 MAPK inhibitor (SB202190) and additionally stimulated with 100 nM of CK2.3 had a significant reduction in the number of cells. In conclusion, we report that ERK1/2 signaling pathway plays a key role in CK2.3 mediated osteoblast differentiation in C2C12 cells.

#### 2.5.2. 200 pM of Smad4 siRNA Induces Significant Reduction in Mineralization Without Affecting the Viability of C2C12 Cells

Smad1/5/8 is one of the most well studied signaling pathway in mediating osteogenesis, however there are not any commercially available pharmacological inhibitors to prevent the activation of Smad1/5/8 molecule. Therefore, we utilized Smad4 siRNA to silence the Smad1/5/8 signaling pathway. Smad4 is a common mediator of Smad1/5/8, which aides in the translocation of Smad1/5/8 into the nucleus, where Smad1/5/8 acts as a transcription factor by directly binding to the DNA and regulating the expression of osteoblast specific genes [51,52]. Furthermore, Smad4 is reported to be essential in regulating osteoblast activity, as its deletion in mature osteoblasts resulted in reduced osteoblast function [64]. Therefore, by employing Smad4 siRNA we effectively prevent the mRNA synthesis of Smad4, thereby Smad1/5/8 is unable to translocate to the nucleus. In our study, C2C12 cells were either left untreated or treated with 200 pM of Smad4 siRNA. The siRNA and the reagent used as the vehicle to transfect the cells are identical to a previously published paper [31] and it is reported that this particular Smad4 siRNA is successful in knocking down 50% of Smad4 [65]. To verify the accuracy of Smad4 siRNA, cells were also treated with 200 pM of control siRNA-A as a positive control. After 24 h of treatment, C2C12 cells were either left unstimulated or further stimulated with 100 nM of CK2.3. Stimulation of C2C12 cells with or without 200 pM of Smad4 siRNA, 200 pM of control siRNA-A, and 100 nM of CK2.3 were repeated for a total of two times during the course of the experiment. At the end, the cells were fixed and stained for calcium deposits using von Kossa assay. Analysis of mineralization was performed using Image J. Our data revealed that cells stimulated with just 100 nM of CK2.3 and cells that were stimulated with 200 pM of control siRNA-A and 100 nM of CK2.3 induced significant mineralization compared to unstimulated cells by 2.36 ± 0.19 and 1.75 ± 0.18 fold respectively. However, cells that were treated with 200 pM of Smad4 siRNA and 100 nM of CK2.3 had a significantly reduced number of mineralization by a fold of 0.91 ± 0.07 (Figure 6A). Cell quantification experiment and MTT reduction assay was conducted to determine if Smad4 siRNA affected the viability of the cells. Our data showed that viability of cells (Figure 6B) and total number of cells (Figure 6C) treated with 200 pM of Smad4 siRNA and 100 nM of CK2.3 did not change compared to unstimulated cells, cells treated with 100 nM of CK2.3, or cells treated with 200 pM of control siRNA-A and 100 nM of CK2.3. Thus, this implies that the reduction of mineralization by CK2.3 in cells stimulated with Smad4 siRNA was due to the inhibition of Smad1/5/8 signaling pathway.

### 2.6. CK2.3 Mediates Osteogenesis via the Smad1/5/8 and ERK1/2 Signaling Pathways in Primary BMSCs Isolated From 4-Month-Old Female C57BL/6J Mice 

Primary BMSCs were isolated from the tibia and femur of 4-month-old female C57BL/6J mice [18,66,67]. C57BL/6J mice are an ideal mice model to study age-related forms of osteoporosis due to their inherent low bone mineral density [68]. BMSCs were either left unstimulated or stimulated with 500 nM of U0126-EtOH (ERK1/2 inhibitor), 200 pM of Smad4 siRNA, and 200 pM of control siRNA-A. The next day, the cells were again either left unstimulated or stimulated with 100 nM of CK2.3. Treatment of cells with signaling inhibitor/siRNAs, followed by CK2.3 stimulation, was repeated for a total of two times over the course of the experiment. Later, the cells were fixed and stained for calcium deposits using von Kossa assay. Our data in primary BMSCs revealed that cells treated with only 100 nM of CK2.3 and cells treated with 200 pM of control siRNA-A and 100 nM of CK2.3 induced significant calcification compared to unstimulated cells by a fold of 3.16 ± 0.34 and 2.26 ± 0.3, respectively. However, cells stimulated with 500 nM of U0126-EtOH (ERK1/2 inhibitor) had significantly reduced amount of calcium deposits compared to CK2.3 treated cells by a fold of 1.31 ± 0.17. Similarly, cells stimulated with 200 pM of Smad4 siRNA had a significantly reduced number of calcium deposits compared to cells stimulated with only CK2.3 and 200 pM of control siRNA-A, by a fold of 1.06 ± 0.14 (Figure 7). Taken altogether, CK2.3 mediates osteoblast differentiation through ERK1/2 and Smad1/5/8 signaling pathways, similar to C2C12 cells.

## 3. Discussion

Osteoporosis is a skeletal disorder and it is associated with loss of bone mass, structure, and integrity over a period of years, thereby increasing the incidence of fractures either at the hip, spine, or wrist. Mortality and morbidity of patients following hip and vertebral fractures is a common consequence of osteoporosis and has a far-reaching impact on the individuals’ quality of life [69]. It is observed that even though women are disproportionately affected by osteoporosis, men are at a higher risk of mortality post hip surgery, at a rate of 36% compared to women at 21%. Mortality rate is seen to be high in the months following hip surgery and peaks at 1 year [70]. We have developed a BMPRIA mimetic peptide called CK2.3 and we have reported about its unique capability to induce calcification and bone growth in in vitro and in vivo studies, respectively [16,17,18,19,33]. However, the mechanism of CK2.3-mediated osteogenesis had been elusive. Here, we undertook the task of characterizing the molecular sequence of events initiated by CK2.3 inducing the differentiation of C2C12 cells and primary BMSCs towards osteoblasts.

Our RT-qPCR data show that *OSX* is significantly upregulated on day 2 and day 3 post-CK2.3 stimulation (Figure 1B) but did not induce a statistically significant increase in *RUNX2* mRNA expression (Figure 1A). It is well known that *RUNX2* is indispensable in the process of osteogenesis, homozygous deletion of *RUNX2* in mice resulted in complete loss of osteoblasts [71,72,73] and haploinsufficiency of *RUNX2* in humans leads to a disease known as cleidocranial dysplasia, which is characterized by defects in bone formation [42,71,74,75]. It also functions by inducing expression of other genes necessary for osteoblast differentiation, such as *Osterix* [72]. *OSX* is required for the differentiation of pre-osteoblasts into fully functioning osteoblasts, by inhibiting the differentiation of MSCs towards chondrogenesis [42,76]. Furthermore, deletion of *OSX* in mouse embryos results in total absence of osteoblasts and bone formation in the embryo stage [77]. It has been reported that *OSX* functions downstream of *RUNX2* during osteoblast differentiation since in *RUNX2*-null mice, the expression of *OSX* was completely abolished, however, deletion of *OSX* in embryos had no effect on the expression of *RUNX2* [77]. However, there are also some conflicting reports that suggest that *RUNX2* is not involved in the induction of *OSX* [78] and BMP2 treatment induced *OSX* expression in *RUNX2*-deficient mesenchymal cells [79]. Collectively, these reports imply that *OSX* expression was induced and activated independent of *RUNX2*.

We show, through immuno-precipitation of BMPRIA, followed by a Western blot, that CK2.3 induces the release of CK2 from BMPRIA starting at 6 h and reaching complete dissociation at 18 h post-stimulation (Figure 2). Simultaneously, CK2.3 colocalized with CK2 starting at 6 h post stimulation (Figure 3), indicating that CK2.3 binds to CK2 and causes it to dissociate from the BMPRIA receptor, thereby activating BMPRIA downstream signaling pathways and osteogenic specific genes. We developed a fluorescent bio-imaging CK2.3-Qdot^®^s probe to track CK2.3 and visualize its interaction with intracellular proteins. In Figure 3, we show that CK2.3-Qdot^®^s interacted with CK2α starting at 6 h and the interaction was maximal at 12 h post stimulation. We further wanted to verify the interaction between CK2.3 and CK2α using an immuno-precipitation method, and also determine whether CK2.3 phosphorylates CK2α protein, but due to the low molecular weight of CK2.3 (3.6 kDa), it has been very difficult to characterize the effect of CK2.3 on protein kinase CK2. 

The BMP2 signaling cascade begins upon ligand binding and formation of hetero-tetrameric complex consisting of BMPRIA and BMPRII, constitutively active BMPRII phosphorylates the highly conserved TTSGSGSG motif (GS domain) present on BMPRIA [52,53,80]. Activated BMPRIA either phosphorylates the immediate downstream molecules Smad1/5/8, and this constitutes the canonical BMP signaling cascade [51,52,53,54]. However, several non-canonical/Smad-independent signaling pathways have also been reported to be involved in osteogenesis such as ERK1/2, p38 MAPK, and Akt1/2/3 [55,56,57,58,59,60,61,62]. Specifically, ERK1/2 and p38 MAPK belong to the mitogen activated protein kinase (MAPK) cascade; MAPKs are proline-directed, serine-threonine kinases. Especially, activation of ERK1/2 signaling is initiated through receptor tyrosine kinases [81], which phosphorylates the Raf-MEK1/2-ERK module [82,83,84,85]. In the case of p38 MAPK, the signaling cascade is composed of MAPKKK-MKK3/4/6-p38MAPK [86,87,88] Lastly, with respect to Akt1/2/3 signaling, activation of receptor tyrosine kinases leads to the recruitment and activation of class I phosphoinositide 3-kinase (PI3K) isoforms to the plasma membrane, ultimately leading to the activation of Akt1/2/3 [89,90]. 

We have previously shown that CK2.3 induces osteogenesis downstream of BMPRIA, independent of BMP2 ligand. However, the function of CK2.3 was not well characterized. Here, we delineated the mechanism of CK2.3. Our Western blot (Figure 4) and signaling inhibition data in C2C12 cells (Figure 5 and Figure 6) and primary BMSCs (Figure 7) reveal that CK2.3 mediates osteogenesis via ERK1/2 and Smad1/5/8 signaling pathways. Interestingly, based on our Western blot data, it appears that CK2.3 mediates the phosphorylation of either only Smad1 or Smad5 signaling molecule (Figure 4A). The role of Smad1/5/8 in mediating osteogenesis is well-documented [22,80,91]. Furthermore, Smad1/5/8 molecules are reported to directly bind to DNA sequences [92] and regulate the expression of target genes known to be involved in osteoblast differentiation [93,94] such *RUNX2* [95] and *Dlx5* [78,96], which then induces the expression of other downstream osteoblastic genes. Even though the role of ERK signaling in mediating osteogenesis is less extensively studied, it is still a critical player [97,98,99,100]. The ERK signaling pathway is reported to mediate *OSX* expression [101] and MAP kinases can also induce the expression of *ALP* and *OCN* [58,59]. 

Our Western blot data (Figure 4) revealed that stimulation of C2C12 cells with 100 nM of CK2.3 activated Smad1/5/8, ERK1/2, and Akt1/2/3 signaling pathways; this can be attributed to the fact that most signaling pathways share common mediators and hence there is extensive cross-talk amongst one another [55]. It has been reported that Ras family of small GTPases, a member of the ERK signaling pathway, interacts [102] and regulates PI3K of Akt pathway [103]. Furthermore, the upregulation in expression of ERK1/2 and Smad1/5/8 molecules by CK2.3 could be due to the involvement of Ras/MEK/ERK pathway in the activation of Smad1. Yue et al. (2000) have shown that in untransformed epithelial cells, inactivation of Ras/MEK pathway greatly affected the phosphorylation of endogenous Smad1 by TGF-β and BMP [104]. Additionally, four consensus ERK phosphorylation sites are reported to be located in Smad1 and these ERK residues are reported to be essential in facilitating the interaction between Smad1-Smad4 complex, nuclear translocation of the complex, and activation of the Smad binding element (SBE) [105].

Taking all the data together, we propose the following mechanism of CK2.3 mediated osteogenesis. In native C2C12 cells, we hypothesize that protein kinase CK2 interacts with BMPRIA (Figure 8A). Stimulation of cells with CK2.3 results in its internalization starting at 6 h through caveolae mediated endocytosis (Figure 8B). At 12 h post stimulation, CK2.3 co-localizes with endogenous CK2α, which then results in the release of CK2 from the receptor BMPRIA (Figure 8C). The dissociation thereby activates BMPRIA downstream signaling pathways such as Smad1/5/8 and ERK1/2, followed by upregulation of osteoblast specific genes like *OSX*, *ALP*, and *OCN* (Figure 8D). This ultimately causes the differentiation of C2C12 cells into osteoblasts. 

## 4. Materials and Methods

### 4.1. CK2.3 Peptide

CK2.3 (1.915 μM) is a custom designed osteogenic peptide synthesized at GenScript (Piscataway, NJ, USA). It is a 29 amino acid long peptide and has a molecular weight of 3655.34 Da. It consists of the CK2 phosphorylation site, which is found on BMPRIA, at amino acids 213–217 (SLKD) incorporated in its sequence, along with the Antennapedia homeodomain signal sequence at the N-terminal of the peptide.

### 4.2. Cell Culture

#### 4.2.1. C2C12 cells (murine myoblast cells) were purchased from American Type

Culture Collection (Manassas, VA, USA). Cells were grown in DMEM (Hy-Clone, Pittsburgh, PA, USA) supplemented with 20% (v/v) heat in-activated FBS (Gemini Bio-products, West Sacramento, CA, USA), and 1% (v/v) penicillin/streptomycin (Hy-clone, Pittsburgh, PA, USA). Phenol-free DMEM media (Hy-Clone, Pittsburgh, PA) supplemented only with 1% (v/v) penicillin/streptomycin (Hy-clone, Pittsburgh, PA), but no FBS. 1X Trypsin (Cellgro, Manassas, VA, USA) was used to detach cells from flasks during sub-culturing.

#### 4.2.2. Isolation of BMSCs from the Femur and Tibia of 4-month-old Female C57BL/6J Mice

Primary BMSCs were isolated by flushing the tibia and femoral marrow compartments of 4-month-old female C57BL/6J mice; purchased from The Jackson Laboratory (Bar Harbor, ME, USA). Our lab has an active IACUC approved protocol (AUP #1194, protocol approved on 2/11/2017) to isolate BMSCs from C57BL/6J mice. All animals for this study were under the direct care and supervision of the Office of Laboratory Animal Medicine at University of Delaware. All animal users complied with IACUC and OLAM guidelines for the care and use of laboratory animals. Collection and culturing of BMSCs were carried out as per the following publication [66]. Cells were cultured in MEM-Alpha Medium (Corning Cellgro, Manassas, VA, USA) supplemented with 20% (v/v) heat in-activated FBS (Gemini Bio-products, West Sacramento, CA, USA), and 1% (v/v) penicillin/streptomycin (Hy-clone, Pittsburgh, PA, USA). Then, 0.25% Trypsin (Cellgro, Manassas, VA, USA) containing 0.02% EDTA (Fisher Chemical, Fair Lawn, NJ, USA) was used to detach the cells from the flasks.

### 4.3. Quantitative Reverse Transcription Polymerase Chain Reaction

C2C12 cells were grown in 35 mm dishes at a cell density of 4.5 × 10^4^ cells/dish for two days in DMEM media containing 20% FBS and 1% penicillin/streptomycin, and at the end of day 2, cells were serum starved with DMEM media supplemented with 1% penicillin/streptomycin, but without FBS for 18 h. Following starvation, cells were either left unstimulated or stimulated with 100 nM of CK2.3 for 6 h, 12 h, 18 h, day 1, day 2, day 3, day 4, and day 5. After each respective time point, total RNA was isolated using Trizol (Ambion life technologies, Carlsbad, CA, USA) method in accordance with Direct-zol™ RNA miniprep manufactures protocol (catalog #R2050, Zymo Research, Irvine, CA, USA). Two-step RT-PCR was performed with 2 μg of RNA obtained, using high-capacity cDNA reverse transcription kit (catalog #4368814, Thermo Fisher Scientific, Waltham, MA, USA). In the second step, cDNA obtained was used for PCR amplification using specific primers. The primers were purchased from Integrated DNA Technologies (Coralville, IA, USA). The primer sequences are as follows: (1) *RUNX-II/p57* (Forward) *TCT GGA AAA AAA AGG AGG GAC TAT G* and *RUNX-II/p57* (Reverse) *GGT GCT CGG ATC CCA AAA GAA* [106,107], (2) *Osterix* (Forward) *GGG TTA AGG GGA GCA AAG TCA GAT* and *Osterix* (Reverse) *CTG GGG AAA GGA GGC ACA AAG AAG* [108,109], (3) *ALP* (Forward) *TCA GGG CAA TGA GGT CAC ATC* and *ALP (Reverse) CAC AAT GCC CAC GGA CTT C* [110] and (4) *Osteocalcin* (Forward) *CTG AGT CTG ACA AAG CCT TC* and *Osteocalcin* (Reverse) *CTG GTC TGA TAG CTC GTC AC* [107]. The primer sequences have been used and verified in the respective publications. RT-qPCR was performed using Fast SYBR™ Green Master Mix per the manufacturers protocol (catalog #4385612, Thermo Fisher Scientific, Waltham, MA, USA) to analyze PCR products or amplicon. *GAPDH* was used as the house-keeping gene, using specific primers for *GAPDH* (Forward) *CAT GGC CTT CCG TGT TCC TA* and *GAPDH* (Reverse) *CCT GCT TCA CCA CCT TCT TGA T* [107]. Fold-change in gene expression was processed using commercially available qbase plus software (Biogazelle, Zwijnaarde, Belgium).

### 4.4. Immunofluorescence Labeling of C2C12 Cells and Analysis

#### 4.4.1. Time-Dependent Release of CK2 from BMPRIA at 12 h and 18 h Post CK2.3 Stimulation

C2C12 cells were grown in 35 mm dishes containing coverslips at a cell density of 1 × 10^4^ cells per well for two days in DMEM media containing 10% FBS and 1% penicillin/streptomycin. Cells were serum starved in DMEM media with only 1% penicillin/streptomycin for 18 h. They were either left unstimulated or treated with 100 nM of CK2.3 for 6 h, 12 h, and 18 h. After each time point, cells were fixed using 4.4% (w/v) paraformaldehyde (Sigma-Aldrich, St. Louis, MO, USA) for 20 min and permeabilized with 0.05% (w/v) saponin (Sigma-Aldrich, St. Louis, MO, USA) diluted in diH_2_O for 10 min on ice. Cells were blocked for non-specific binding using 3% BSA (Fischer Scientific, Hampton, NH, USA) diluted in 1 × PBS at room temperature for 1 h. The cells were then labeled for CK2α (Santa Cruz Biotechnology, Dallas, TX, USA) and BMPRIA (Santa Cruz Biotechnology, Dallas, TX, USA) diluted in 3% BSA at room temperature for 1 h. Following this incubation, the cells were washed three times with 1X PBS for five minutes each and then labeled for the corresponding secondary antibodies donkey anti-rabbit IgG H&L (Alexa Fluor 568, Cambridge, MA, USA) and chicken anti-goat IgG H&L (Alexa Fluor 488, Cambridge, MA, USA) for 1 h at room temperature. The cells were washed three times with 1X PBS for five minutes each and then stained for the nucleus using Hoechst (Sigma-Aldrich, St. Louis, MO, USA) for 2.5 min. The coverslips were mounted on clean, labeled slides using Airvol and the slides were imaged using Zeiss LSM 710 at 63×/1.4 Plan-Aprochromat oil objective.

#### 4.4.2. Co-Localization of CK2.3-Qdot^®^s with Endogenous CK2α Protein Within C2C12 Cells

C2C12 cells were grown in 35 mm dishes containing 18X18 mm square coverslips at a cell density of 1 × 10^4^ cells/well for two days in DMEM media containing 20% FBS and 1% penicillin/streptomycin. At the end of the second day, cells were serum starved with phenol-free DMEM media supplemented only with 1% penicillin/streptomycin for 18 h. C2C12 cells were either left unstimulated or stimulated with 100 nM of CK2.3-Qdot^®^s for 6 h, 12 h, and 18 h. After each respective time point, cells were fixed using 4.4% (w/v) paraformaldehyde (Sigma-Aldrich, St. Louis, MO, USA) for 20 min and permeabilized with 0.05% (w/v) saponin (Sigma-Aldrich, St. Louis, MO, USA), diluted in diH2O, and incubated for 10 min on ice. Cells were blocked with 3% (w/v) protease-free BSA (Fisher Scientific, Hampton, NH, USA) diluted in 1 × PBS pH 7.4 (Corning, Manassas, VA, USA), at room temperature for 1 h. Following blocking, cells were labeled for CK2α using 100 μL (2 μg/mL) of rabbit polyclonal IgG casein kinase IIα antibody (catalog #sc-9030, Santa Cruz Biotechnology, Dallas, TX, USA) diluted in 3% protease-free BSA in the ratio of 1:100, at room temperature for 1 h. The primary antibody was then stained against using 100 μL (1 μg/μL) of fluorescently tagged-secondary antibody, donkey anti-rabbit IgG H&L (Alexa Fluor^®^ 568, catalog #ab175470, Abcam, Cambridge, MA, USA) diluted in the ratio of 1:500 in 3% protease-free BSA, at room temperature for 1 h. Later, the nucleus was stained using 100 μL (0.5 ng/mL) of Hoechst (catalog #23491-45-4, Sigma-Aldrich, St. Louis, MO, USA). The coverslips were then mounted on the slides using Airvol. The specificity of donkey anti-rabbit Alexa Fluor^®^ 568 was determined by labeling unstimulated cells with only the fluorescent secondary antibody. Slides were imaged using Zeiss LSM 710 at 63×/1.4 Plan-Apochromat oil objective.

### 4.5. Western Blot

#### Detecting the Expression Pattern of BMPRIA Downstream Signaling Pathways by CK2.3

C2C12 cells were grown in 100 mm dishes at a cell density of 39.25 × 104 cells/dish for two days in DMEM media containing 20% FBS and 1% penicillin/streptomycin and at the end of the second day, cells were serum starved with DMEM media supplemented only with 1% penicillin/streptomycin for 18 h. Following starvation, cells were either left unstimulated or stimulated with 100 nM of CK2.3 for 6 h, 12 h, 18 h, and days 1–5. After each respective time point, cells were washed with ice cold 1X PBS (Corning, Manassas, VA, USA) and later scraped off the dish and centrifuged at 2500× *g* for 5 min. The pellet was extracted using NP-40 lysis buffer (150 mM NaCL (Fisher Chemical, Fair Lawn, NJ, USA), 50mM Tris (Fisher Bioreagents, Fair Lawn, NJ, USA), 1.0% NP-40 (AbcamBiochemicals, Cambridge, United Kingdom), pH 8) in the presence of protease and phosphatase inhibitor cocktail (Sigma-Aldrich, St. Louis, MO, USA). After 30 min on ice, cells were sonicated (30 s × 3). After lysing the cells, the cell debris was removed by centrifugation at 13,000× *g* for 20 min and the supernatant containing the cytoplasmic extract was transferred to a new tube [111]. Protein content of the cell lysates were determined using the BCA assay (Thermo Scientific, Rockford, IL, USA) and were normalized to equal protein content. Protein extracts were analyzed using SDS-polyacrylamide gel electrophoresis followed by Western blot. The samples were analyzed for Smad1/5/8 (N-18) (catalog #sc-6031-R, Santa Cruz Biotechnology, Dallas, TX, USA), Phospho-Smad1(Ser463/465)/Smad5(Ser463/465)/Smad8(S426/428) (catalog #9511S, Cell Signaling Technology, Danvers, MA, USA), p44/42 MAPK (ERK1/2) (catalog #9102S, Cell Signaling Technology, Danvers, MA, USA), Phospho-p44/42 MAPK (ERK1/2) (Thr202/Tyr204) (catalog #9101S, Cell Signaling Technology, Danvers, MA, USA), p38 MAPK (D13E1) XP (catalog #8690T, Cell Signaling Technology, Danvers, MA, USA), Phospho-p38 MAPK alpha (Thr180, Tyr182) (catalog #MA5-5182, ThermoFisher Scientific, Waltham, MA, USA), Phospho-Akt (Ser473) (D9E) XP (catalog #4060S, Cell Signaling Technology, Danvers, MA, USA), Akt1/2/3 (5C10) (catalog #sc-81434, Santa Cruz Biotechnology, Dallas, TX, USA), and Beta actin (catalog #sc-20536-1-AP, Rosemont, IL, USA) was used as the loading control. The experiments were repeated at least three times.

### 4.6. Immunoprecipitation

C2C12 cells were grown in 6 well plates at 1.0 × 10^5^ cells/mL for two days in DMEM with 10% FBS and 1% penicillin/streptomycin. After the second day, the cells were serum starved for 18 h. They were then stimulated with 100 nM of CK2.3 or left unstimulated for 6 h, 12 h, and 18 h. After the designated time point the cells were lysed following the same protocol as outlined above in the Western blot section. PureProteome Protein G magnetic beads (EMD Millipore, Burlington, MA, USA) were incubated and washed three times with lysis buffer (NP-40), centrifuging at 14,000 rpm for five minutes at 4 °C. The beads were then incubated with BMPRIA (rabbit polyclonal, Santa Cruz, Dallas, TX, USA) for 4 h at 4 °C. After lysates were collected, they were incubated with BMPRIA/magnetic beads overnight at 4 °C. The beads/BMPRIA were then washed three times with lysis buffer (NP-40), centrifuging at 14,000 rpm for five minutes at 4 °C. The IP lysates were analyzed using SDS-polyacrylamide gel electrophoresis, followed by a Western blot. They were detected for BMPRIA (rabbit polyclonal, Santa Cruz, Dallas, TX, USA), and CK2β (rabbit polyclonal, Santa Cruz, Dallas, TX, USA).

### 4.7. siRNA Transfection

C2C12 cells were grown in a 24-well plate at a cell density of 1 × 10^4^ cells/well for two days in DMEM media containing 20% FBS and 1% penicillin/streptomycin and at the end of the second day, cells were serum starved for 18 h with DMEM media containing only 1% penicillin/streptomycin. Cells were then either left unstimulated or stimulated with 200 pM of Smad4 siRNA (m) (catalog #sc-29485, Santa Cruz Biotechnology, Dallas, TX, USA) and 200 pM control siRNA-A (catalog #sc-37007, Santa Cruz Biotechnology, Dallas, TX, USA) for 24 h. Cells were transfected with siRNA using Turbofect transfection reagent (catalog #R0531, Thermo Scientific, Waltham, MA, USA) as per the protocol. The next day, cells were either left untreated or treated with 100 nM of CK2.3. Transfection of cells with siRNA, followed by 100 nM of CK2.3 was repeated again on day 4. On the sixth day, cells were either fixed using 4.4% (w/v) paraformaldehyde (Sigma-Aldrich, St. Louis, MO, USA) and stained for mineralization using von Kossa assay or viability of the cells was determined using MTT reduction assay.

### 4.8. von Kossa Assay

von Kossa assay is a biochemical test used to quantify the amount of mineralization. In this assay, the silver cations of the silver nitrate solution react with phosphates and carbonates present in the calcium deposits, which are then reduced to black metallic silver stain when exposed to UV light [63,112].

#### 4.8.1. Inhibition of BMPRIA Downstream Signaling Pathways in C2C12 Cells

C2C12 cells were grown in a 24-well plate at a cell density of 1 × 10^4^ cells/well for two days in DMEM media containing 20% FBS and 1% penicillin/streptomycin and at the end of the second day, cells were serum starved for 18 h with DMEM media containing 1% penicillin/streptomycin and without FBS. Cells were then either left unstimulated or stimulated with 500 nm, 1 μM, and 5 μM of U0126-EtOH (MEK1/2 inhibitor, catalog #S1102, Selleckchem, Houston, TX, USA). U0126-EtOH is an inhibitor of intracellular Raf/MEK/ERK signaling pathway. One set of C2C12 cells were either left unstimulated or stimulated with 500 nm, 1 μM, and 5 μM of MK-2206 2HCl (Akt1/2/3 inhibitor, catalog #S1078, Selleckchem, Houston, TX, USA). The Akt inhibitor, MK-2206 2HCl is reported to inhibit autophosphorylation of Akt at T308 and S473, as well as prevent Akt-mediated activation of downstream molecules. Also, one more set of cells were either left unstimulated or stimulated with 500 nm, 1 μM, 5 μM, and 10 μM of SB202190 (FHPI) (p38 MAPK inhibitor, catalog #S1077, Selleckchem, Houston, TX, USA). SB202190 (FHPI) is a potent inhibitor of p38 MAPK and it functions by targeting p38α/β. The above-mentioned inhibitors have been previously used to inhibit the respective signaling pathways in C2C12 cells [113,114,115]. The cells were stimulated with the reported concentrations of the inhibitors on day 1 and then again on day 4. Cells were also stimulated with 100 nM of CK2.3 on day 2 and day 5. Further, cells were supplemented with DMEM media containing 10% FBS and 1% penicillin/streptomycin on day 3 and day 6. After 6 complete days of stimulation, cells were fixed using 4.4% (w/v) paraformaldehyde (Sigma-Aldrich, St. Louis, MO, USA) for 20 min and washed with ice cold 1X PBS pH 7.4. 5% (w/v) silver nitrate (Chem-Impex international, Wood Dale, IL, USA) dissolved in diH_2_O, which was applied to each well, and the plate was put under the UV light for 1 h. Cells were then washed with diH_2_O and were allowed to dry overnight. Twelve random images of the cells were taken per well using Zeiss Axiovert 10 microscope at 5×/.12 The achrostigmat objective and images were analyzed using ImageJ software (NIH, Bethesda, MD, USA). Images were first converted to 8 bit and threshold was set to the positive control. The same threshold was used for all treatments. Mineralized areas were quantified using the analyzing particles plugin function of ImageJ.

#### 4.8.2. Inhibition of Smad1/5/8 and ERK1/2 Signaling Pathways in Primary BMSCs

Primary BMSCs isolated from the tibia and femur of 4-month-old female C57BL/6J mice were grown in a 24-well plate at a cell density of 1 × 10^4^ cells/well in MEM alpha medium containing 20% FBS and 1% penicillin/streptomycin. Once the cells reach 90% confluency, cells were then either left unstimulated or stimulated with 500 nM of U0126-EtOH (ERK inhibitor, catalog #S1102, Selleckchem, Houston, TX, USA), 200 pM of Smad4 siRNA (m) (catalog #sc-29485, Santa Cruz Biotechnology, Dallas, TX, USA), and 200 pM of control siRNA-A (catalog #sc-37007, Santa Cruz Biotechnology, Dallas, TX, USA) on day 1 and then again on day 4. Cells were also stimulated with 100 nM of CK2.3 on day 2 and day 5. After 6 complete days of stimulation, cells were fixed using 4.4% (w/v) paraformaldehyde (Sigma-Aldrich, St. Louis, MO, USA) for 20 min and washed with ice cold 1X PBS pH 7.4. 5% (w/v) silver nitrate (Chem-Impex international, Wood Dale, IL, USA) dissolved in diH_2_O, which was applied to each well and the plate was put under the UV light for 1 h. Cells were then washed with diH_2_O and were allowed to dry overnight. Twelve random images of the cells were taken per well using Zeiss Axiovert 10 microscope at 5×/.12 The achrostigmat objective and images were analyzed using ImageJ software (NIH, Bethesda, MD, USA). Images were first converted to 8 bit and threshold was set to the positive control. The same threshold was used for all treatments. Mineralized areas were quantified using the analyzing particles plugin function of ImageJ. 4.9. MTT reduction assay.

MTT reduction assay is a marker for viable cell metabolism. Viable cells with active metabolism reduce MTT into formazan (a purple colored byproduct), however, when cells die, they lose the ability to convert MTT into formazan. The quantity of formazan is linearly proportional to the number of viable cells and measured by changes in absorbance at 570 nm using a spectrophotometer [116]. C2C12 cells were grown in a 24-well plate at a cell density of 1 × 10^4^ cells/well for two days in DMEM media containing 20% FBS and 1% penicillin/streptomycin, and at the end of the second day, cells were serum starved for 18 h with DMEM media containing only 1% penicillin/streptomycin. Cells were then either left unstimulated or stimulated with 500 nM of U0126-EtOH (ERK1/2 inhibitor, catalog #S1102), 5 μM of MK-2206 2HCl (Akt1/2/3 inhibitor, catalog #S1078), 10 μM of SB202190 (FHPI) (p38 MAPK inhibitor, catalog #S1077), 200 pM of Smad4 siRNA (m) (catalog #sc-29485), and 200 pM of control siRNA-A (catalog #sc-37007 on day 1 and day 3. Cells were further treated with 100 nM of CK2.3 on day 2 and day 5, cells were also supplemented with DMEM media containing 10% FBS and 1% penicillin/streptomycin on days 4 and 5 of the experiment. On the sixth day, viability of the cells following treatment with inhibitors and CK2.3 was determined using MTT (3-(4,5-Dimethylthiazol-2-yl)-2,5-Diphenyltetrazolium Bromide) assay per the manufacturer’s protocol (catalog #M6494, Thermo Fisher Scientific, Waltham, MA, USA).

### 4.9. Statistical Data Analaysis

All the results are presented as mean ± standard error of the mean (SEM). All the experiments were repeated at least three times, outliers were removed from the data based on Chauvenet’s criterion [117], and data was analyzed using the single factor analysis of variance (ANOVA), followed by Tukey-Kramer post hoc statistical test at a 95% confidence level.

## Figures and Tables

**Figure 1 ijms-20-02500-f001:**
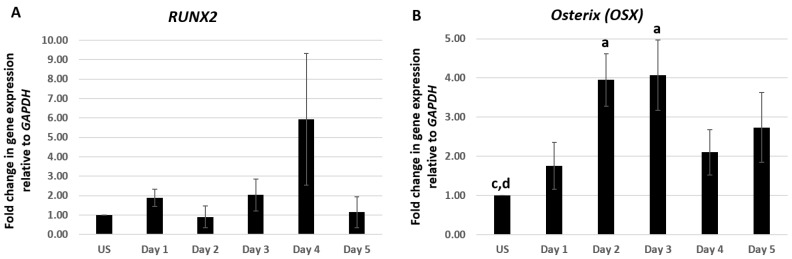
CK2.3 stimulation led to the upregulation of *Osterix, Alkaline Phosphatase,* and *Osteocalcin* genes in C2C12 cells. C2C12 cells were treated with 100 nM of CK2.3 and expression of (**A**) *RUNX2,* (**B**) *OSX*, (**C**) *ALP*, and (**D**) *OCN* were analyzed using RT-qPCR over the course of 5 days. *GAPDH* was used as the house-keeping gene. Data (*n* = 4) was normalized to unstimulated cells and analyzed using one-way anova and Tukey-Kramer post hoc statistical test (*p* < 0.05). a = statistically significant difference to unstimulated, b = statistically significant difference to Day 1, c = statistically significant difference to Day 2, d = statistically significant difference to Day 3, e = statistically significant difference to Day 4, and f = statistically significant difference to Day 5.

**Figure 2 ijms-20-02500-f002:**
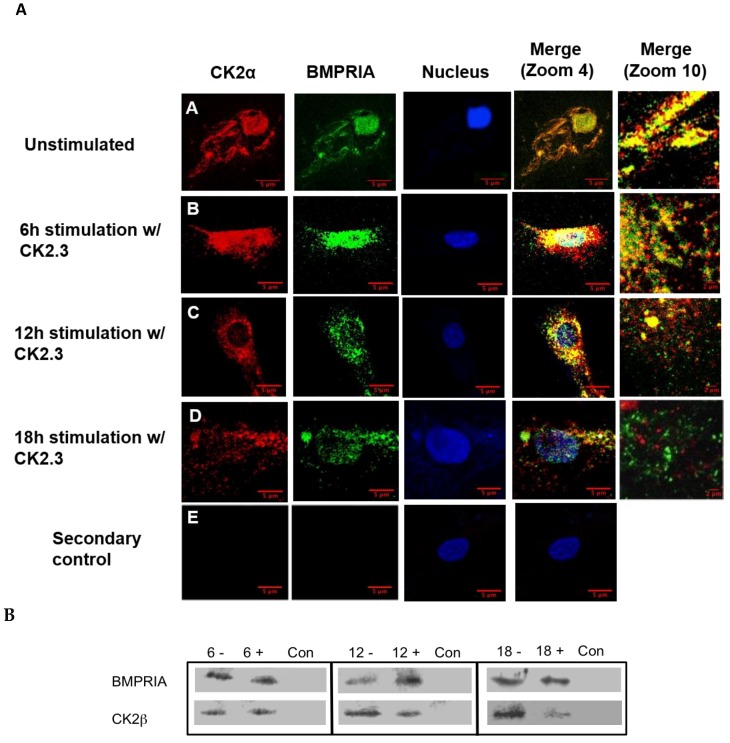
Time-dependent release of CK2 from BMPRIA at 12 h and 18 h post CK2.3 stimulation. (**A**) Visual analysis of the effect of CK2.3 on the interaction between endogenous BMPRIA and CK2α. Confocal images of fixed C2C12 cells that were either (A) unstimulated (-) or stimulated with 100 nM of CK2.3 (+) for (B) 6 h, (C) 12 h, and (D) 18 h. The images depict the interaction between endogenous BMPRIA (green) and CK2α (red) within the cell at different time intervals. The nucleus of the cells is depicted in blue. (E) In the secondary control, unstimulated cells were stained using only the fluorescent secondary antibody to determine their specificity, and lack of staining in the cells shows that the antibody was specific against the antigen. (**B**) Immuno-precipitation of BMPRIA in C2C12 cells, not stimulated (-) or stimulated (+) with 100 nM of CK2.3 for 6 h, 12 h, and 18 h; followed by a co-immuno-precipitation for CK2β that showed reduced interaction of CK2β with BMPRIA at 12 h and 18 h post-stimulation. Con represents the negative control of the immuno-precipitation, where lysis buffer was used instead of cell lysate.

**Figure 3 ijms-20-02500-f003:**
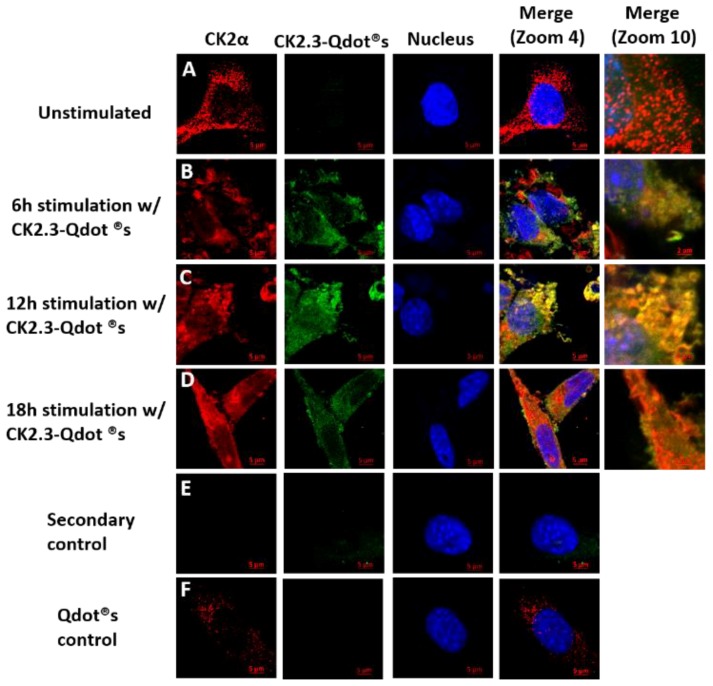
Time-dependent co-localization of CK2 with CK2.3-Qdot^®^s at 6 h, 12 h, and 18 h post stimulation. C2C12 cells were either (**A**) unstimulated or stimulated with 100 nM of CK2.3-Qdot^®^s for (**B**) 6 h, (**C**) 12 h, and (**D**) 18 h. Cells were fixed, stained for CK2α and nucleus of the cell after each respective time points, and images were taken using the LSM 710 confocal microscopy. The images depict the interaction between endogenous CK2α (red) and CK2.3-Qdot^®^s (green) within the cell at different time intervals. The nuclei of the cells are depicted in blue. (E) In secondary control, unstimulated cells were stained using only the fluorescent secondary antibody to determine their specificity, and lack of staining in the cells implies that the antibody is specific against the antigen. (F) In Qdot^®^s control, cells were treated with Qdot^®^s fraction and lack of Qdot^®^s signal in the cells indicate that the interaction between CK2α and CK2.3-Qdot^®^s is driven by CK2.3.

**Figure 4 ijms-20-02500-f004:**
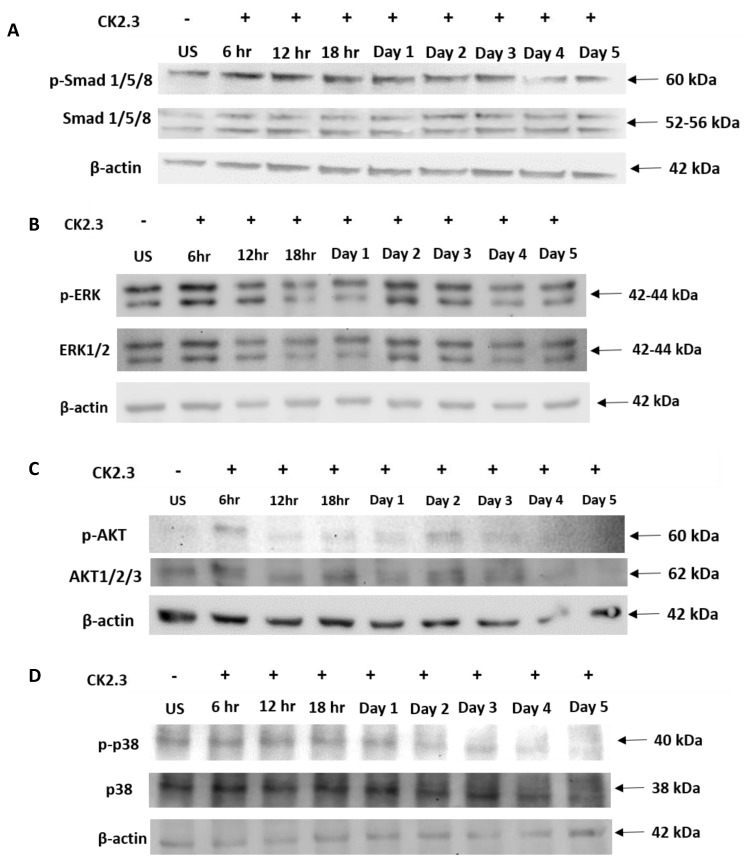
Smad1/5/8, ERK1/2, and Akt1/2/3 signaling pathways are upregulated following stimulation with 100 nM of CK2.3 in C2C12 cells. C2C12 cells were either left unstimulated or stimulated with 100 nM of CK2.3 for 6 h, 12 h, and 18 h over days 1–5 and, after each respective time points, protein was extracted, normalized, and analyzed using SDS-PAGE, followed by Western blot. (**A**) p-Smad1/5/8 and total Smad1/5/8 expression gradually increased with CK2.3 treatment. (**B**) p-ERK expression was elevated and total ERK1/2 expression gradually increased through the 5-day experiment. Similarly, (**C**) p-Akt1/2/3 expression increased with CK2.3 treatment, however, (**D**) p38 MAPK expression stayed relatively constant. β-actin was used as the loading control. An absolute timing of the CK2.3 response was variable over the 5-day experiment. Thus, we did not include a quantitative representation of the Western blots.

**Figure 5 ijms-20-02500-f005:**
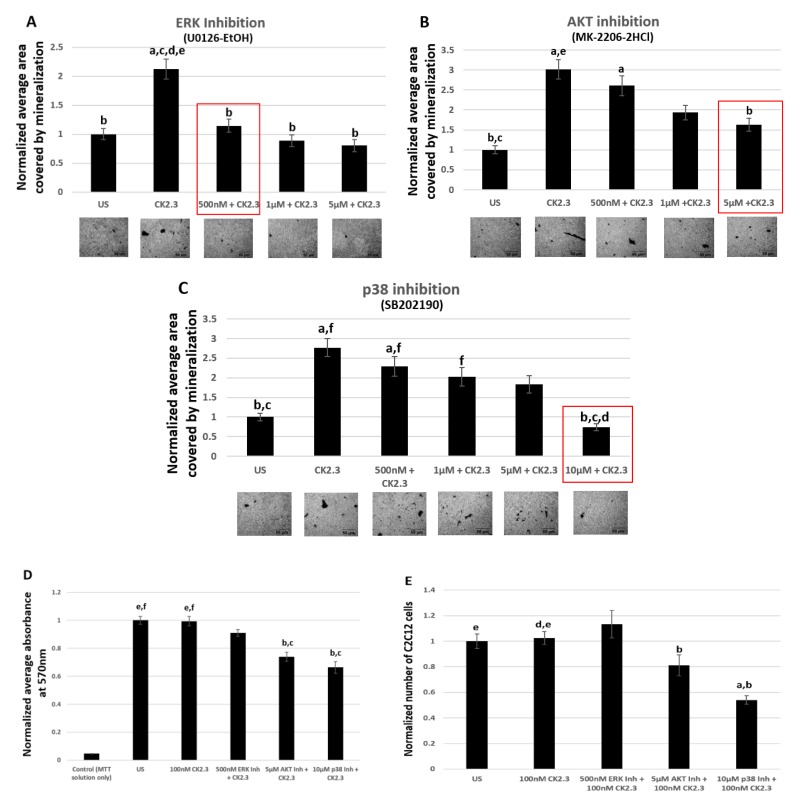
500 nM of ERK1/2 inhibitor (U0126-EtOH) induces significant reduction in mineralization without affecting the viability of C2C12 cells. Mineralization levels in unstimulated C2C12 cells and cells treated with (**A**) 500 nM, 1 μM, and 5 μM of U0126-EtOH (ERK 1/2 inhibitor), (**B**) 500 nM, 1 μM, and 5 μM of MK-2206-2HCl (Akt inhibitor) and (**C**) 500 nM, 1 μM, 5 μM, and 10 μM of SB202190 (p38 inhibitor); followed by stimulation with 100 nM of CK2.3 was determined using von Kossa assay. (**D**) Viability of cells and (**E**) total number of cells after treatment with 500 nM of U0126-EtOH, 5 μM of MK-2206-2HCl, and 10 μM of SB202190 followed by stimulation with 100 nM of CK2.3 was determined using MTT assay and ImageJ analysis, respectively. Data (*n* = 3) was normalized to unstimulated cells and analyzed using one-way anova and Tukey-Kramer post hoc statistical test (*p* < 0.05). (**A**) a = statistically significant difference to unstimulated, b = statistically significant difference to 100 nM CK2.3, c = statistically significant difference to 500 nM ERK1/2 Inh + 100 nM CK2.3, d = statistically significant difference to 1 μM ERK1/2 Inh + 100 nM CK2.3, and e = statistically significant difference to 5 μM ERK1/2 Inh + 100 nM CK2.3. (**B**) a = statistically significant difference to unstimulated, b = statistically significant difference to 100 nM CK2.3, c = statistically significant difference to 500 nM Akt Inh + 100 nM CK2.3, d = statistically significant difference to 1 μM Akt Inh + 100 nM CK2.3, and e = statistically significant difference to 5 μM Akt Inh + 100 nM CK2.3. (**C**) a = statistically significant difference to unstimulated, b = statistically significant difference to 100 nM CK2.3, c = statistically significant difference to 500 nM p38 Inh + 100 nM CK2.3, d = statistically significant difference to 1 μM p38 Inh + 100 nM CK2.3, e = statistically significant difference to 5 μM p38 Inh + 100 nM CK2.3, and f = statistically significant difference to 10 μM p38 Inh + 100 nM CK2.3. (**D**) a = statistically significant difference to control, b = statistically significant difference to unstimulated, c = statistically significant difference to 100 nM CK2.3, d = statistically significant difference to 500 nM ERK1/2 Inh + 100 nM CK2.3, e = statistically significant difference to 5 μM Akt Inh + 100 nM CK2.3, and f = statistically significant difference to 10 μM p38 Inh + 100 nM CK2.3. (**E**): a = statistically significant difference to unstimulated, b = statistically significant difference to 100 nM CK2.3, c = statistically significant difference to 500 nM ERK1/2 Inh + 100 nM CK2.3, d = statistically significant difference to 5 μM Akt Inh + 100 nM CK2.3, and d = statistically significant difference to 10 μM p38 Inh + 100 nM CK2.3.

**Figure 6 ijms-20-02500-f006:**
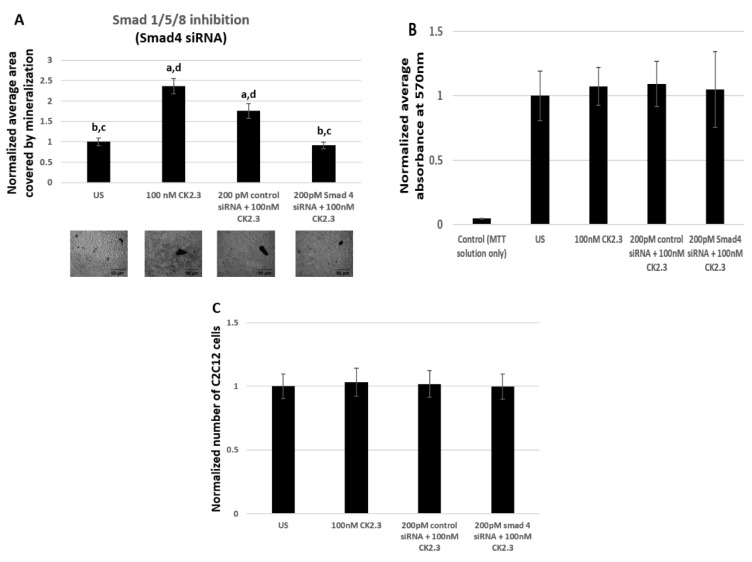
200 pM of Smad4 siRNA induces significant reduction in mineralization without affecting the viability of C2C12 cells. (**A**) Mineralization levels in unstimulated C2C12 cells and cells treated with 200 pM of control siRNA and 200 pM of Smad4 siRNA, followed by stimulation with 100 nM of CK2.3, was determined using von Kossa assay. (**B**) Viability of cells and (**C**) total number of cells after treatment with 200 pM of control siRNA and 200 pM of Smad4 siRNA followed by stimulation with 100 nM of CK2.3 was determined using MTT assay and imageJ software, respectively. Data (*n* = 3) was normalized to unstimulated cells and analyzed using one-way anova and Tukey-Kramer post hoc statistical test (*p* < 0.05). (**A**) and (**C**) a = statistically significant difference to unstimulated, b = statistically significant difference to 100 nM CK2.3, c = statistically significant difference to 200 pM control siRNA + 100 nM CK2.3, and d = statistically significant difference to 200 pM Smad4 siRNA + 100 nM CK2.3. (**B**) a = statistically significant difference to control, b= statistically significant difference to unstimulated, c = statistically significant difference to 100 nM CK2.3, d = statistically significant difference to 200 pM control siRNA + 100 nM CK2.3, and e = statistically significant difference to 200 pM Smad4 siRNA + 100 nM CK2.3.

**Figure 7 ijms-20-02500-f007:**
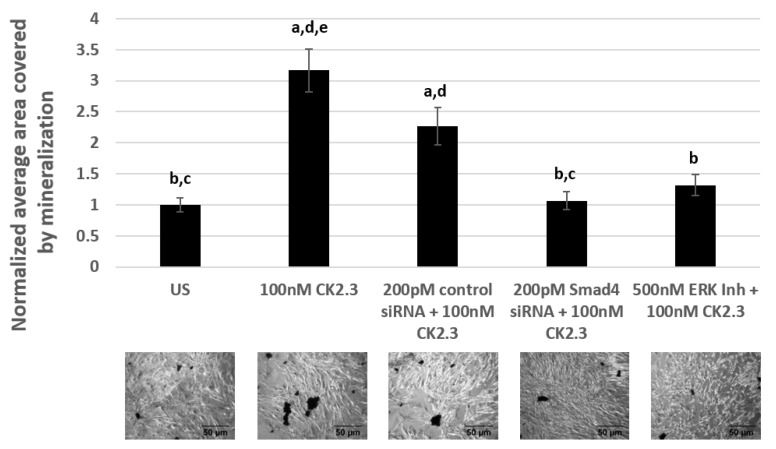
CK2.3 mediates osteogenesis via Smad1/5/8 and ERK1/2 signaling pathways in primary BMSCs isolated from 4-month-old female C57BL/6J mice. Primary BMSCs were either left unstimulated or stimulated with 200 pM of control siRNA, 200 pM of Smad4 siRNA, and 500 nM of U0126-EtOH (ERK1/2 inhibitor) for 24 h, followed by stimulation with 100 nM of CK2.3. Treatment of cells with signaling inhibitor/siRNAs, followed by CK2.3 stimulation, was repeated for a total of two times over the course of a 6-day experiment. Later, the cells were fixed and stained for calcium deposits using von Kossa assay. Data (*n* = 4) was normalized to unstimulated cells and analyzed using one-way anova and Tukey-Kramer post hoc statistical test (*p* < 0.05). a = statistically significant difference to unstimulated, b = statistically significant difference to 100 nM CK2.3, c = statistically significant difference to 200 pM control siRNA + 100 nM CK2.3, d = statistically significant difference to 200 pM Smad4 siRNA + 100 nM CK2.3, and e = statistically significant difference to 500 nM ERK1/2 Inh + 100 nM CK2.3.

**Figure 8 ijms-20-02500-f008:**
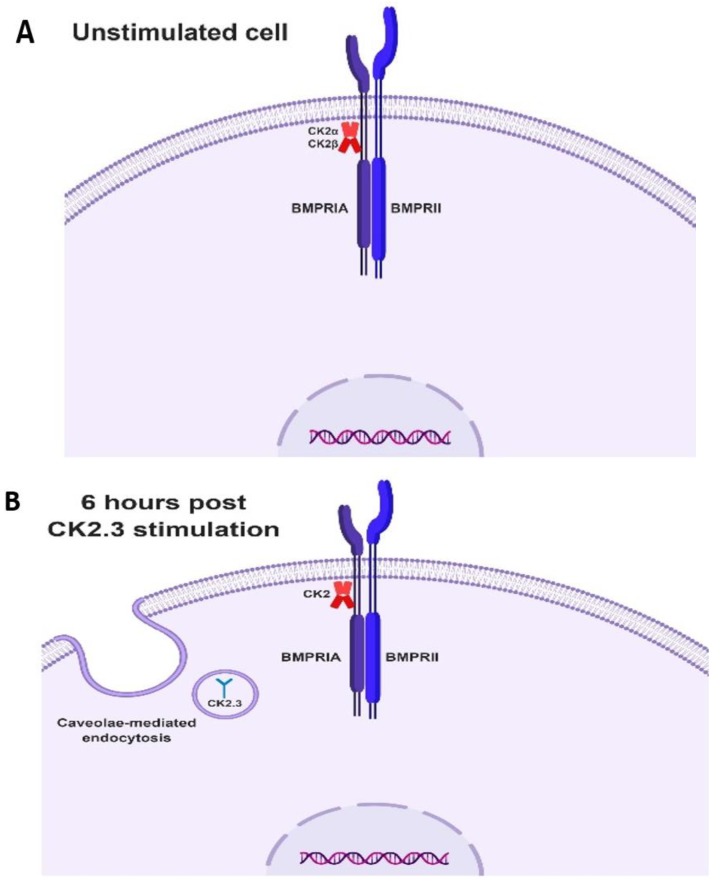
Proposed mechanism of CK2.3 mediated osteogenesis. **(A**) In unstimulated cells, protein kinase CK2 is hypothesized to be interacting with the receptor BMPRIA. (**B**) Stimulation of cells with CK2.3 results in its internalization starting at 6 h via caveolae-mediated endocytosis. (**C**) CK2.3 then interacts with endogenous CK2, causing its release from receptor BMPRIA. (**D**) Thereby activating BMPRIA downstream signaling pathways and genes.

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
