# Peer review of "Mechanism of CK2.3, a Novel Mimetic Peptide of Bone Morphogenetic Protein Receptor Type IA, Mediated Osteogenesis"

_ijms, 2019, doi:10.3390/ijms20102500_

Reviewer 1 Report

In this article the authors aimed to determine the mechanism of CK2.3 in the regulation of osteogenesis. Using a newly developed fluorescent CK2.3 analog, they showed that CK2.3 induces osteogenesis and bone formation in vivo and in vitro through releasing CK2 from BMPRIA. They further confirmed that the mechanism mediating this process involved the activation of BMPRIA downstream signaling pathways Smad1/5/8 and ERK1/2, followed by the upregulation of osteoblast specific genes OSX, ALP, and OCN.

The manuscript was well written and easy to follow. Subject definitions and methods were described clearly. The topic of this paper was relevant to the field of this journal. I have only a few suggestions for improvement.

1)    Please clarify the statistical analysis method in “Materials and Methods”.

2)    What do the letters “a,b,c,d…..” mean in fig1,  fig5, fig6? Please correct them with standard statistical symbols to mark the significant differences in the figures. And also add the information of sample numbers and statistical analysis method in each figure legend.

3)    Line 234 figure 4A, why the p-Smad 1/5/8 has only one band while the total Smad1/5/8 has two bands?

4)     The authors concluded that the binding of CK2.3 with CK2a resulted in the dissociation of CK2 from BMPRIA which thereby activated BMPRIA downstream signaling pathways such as Smad1/5/8 and ERK1/2. According to the results of fig2, the dissociation of CK2 from BMPRIA started at 12 hours post CK2.3 stimulation, however, the expression of  p-Smad1/5/8,p-ERK and p-AKT upregulated at 6 hours post CK2.3 stimulation. How to explain this?

5)      Please use standard unit symbols. For example, in line 317, 329,333, 335,341,346,353…please correct rM with pM.

6)        Please correct “in-vitro” and “in-vivo” to “in vitro” and “in vivo” throughout the manuscript.

7)        Please correct the format of reference 20 in line 798-800.

I recommend acceptance of this paper after minor revision.

Author Response

We appreciate the opportunity to revise our manuscript for reconsideration. We also want to thank the reviewer for the insightful comments on our manuscript. We took every comment seriously and revised the manuscript accordingly. We have attached our rebuttal and addressed the reviewers’ comments and suggestions point-by-point.

Please let us know if we can provide any more information.

Sincerely,

Vrathasha.

Reviewer 2 Report

This is well conducted study. The following concerns should be replied before publication.

1.          Did the authors evaluate the mRNA expression of RUNX2?

2.          Please provide quantitative result of figure 4.

3.          Quantification of von Kossa stain by image J is not adequate. Please perform Alizarin Red S stain and quantified by dissolving the cell-bound Alizarin Red S in 10% acetic acid and then was quantified spectrophotometrically. (may cite : 1. Lin SY, Kang L, Wang CZ, Huang HH, Cheng TL, Huang HT, Lee MJ, Lin YS, Ho ML, Wang GJ, Chen CH*. (-)-Epigallocatechin-3-Gallate (EGCG) Enhances Osteogenic Differentiation of Human Bone Marrow Mesenchymal Stem Cells. Molecules 2018, 23, 3221. 2.        Tai IC, Wang YH, Chen CH, Chuang SC, Chang JK, Ho ML. Simvastatin enhances Rho/actin/cell rigidity pathway contributing to mesenchymal stem cells' osteogenic differentiation. Int J Nanomedicine. 2015 Sep 21;10:5881-94. 3. Tsai YH, Lin KL, Huang YP, Hsu YC, Chen CH, Chen Y, Sie MH, Wang GJ, Lee MJ. Suppression of ornithine decarboxylase promotes osteogenic differentiation of human bone marrow-derived mesenchymal stem cells. FEBS Lett. 2015 Jul 22;589(16):2058-65. 4. Chuang SC, Chen CH, Fu YC, Tai IC, Li CJ, Chang LF, Ho ML, Chang JK. Estrogen receptor mediates simvastatin-stimulated osteogenic effects in bone marrow mesenchymal stem cells. Biochem Pharmacol. 2015 Dec 1;98(3):453-64.)

4.          Line 365: Please provide the reference of “Primary BMSCs were isolated from the tibia and femur of 4-month-old female C57BL/6J mice.”.

5.          The manuscript is too wordy. Please make it succient.

Author Response

(The authors gave the same response as above.)
